# Synthesis and Properties of Novel Polyurethanes Containing Long-Segment Fluorinated Chain Extenders

**DOI:** 10.3390/polym10111292

**Published:** 2018-11-21

**Authors:** Jia-Wun Li, Hsun-Tsing Lee, Hui-An Tsai, Maw-Cherng Suen, Chih-Wei Chiu

**Affiliations:** 1Department of Materials Science and Engineering, National Taiwan University of Science and Technology, Taipei 10607, Taiwan; a12352335@gmail.com; 2Department of Materials Science and Engineering, Vanung University, Jongli, Taoyuan 32061, Taiwan; htlee@mail.vnu.edu.tw; 3R&D Center for Membrane Technology, Department of Chemical Engineering, Chung Yuan University, Chungli District, Taoyuan 32023, Taiwan; huian@cycu.edu.tw; 4Department of Fashion Business Administration, LEE-MING Institute of Technology, No. 22, Sec. 3, Tailin. Rd., New Taipei 24305, Taiwan

**Keywords:** fluorinated polyurethanes, chain extender, curve fitting technique, nuclear magnetic resonance, hydrogen bond

## Abstract

In this study, novel biodegradable long-segment fluorine-containing polyurethane (PU) was synthesized using 4,4′-diphenylmethane diisocyanate (MDI) and 1H,1H,10H,10H-perfluor-1,10-decanediol (PFD) as hard segment, and polycaprolactone diol (PCL) as a biodegradable soft segment. Nuclear magnetic resonance (NMR) was used to perform ^1^H NMR, ^19^F NMR, ^19^F–^19^F COSY, ^1^H–^19^F COSY, and HMBC analyses on the PFD/PU structures. The results, together with those from Fourier transform infrared spectroscopy (FTIR), verified that the PFD/PUs had been successfully synthesized. Additionally, the soft segment and PFD were changed, after which FTIR and XPS peak-differentiation-imitating analyses were employed to examine the relationship of the hydrogen bonding reaction between the PFD chain extender and PU. Subsequently, atomic force microscopy was used to investigate the changes in the microphase structure between the PFD chain extender and PU, after which the effects of the thermal properties between them were investigated through thermogravimetric analysis, differential scanning calorimetry, and dynamic mechanical analysis. Finally, the effects of the PFD chain extender on the mechanical properties of the PU were investigated through a tensile strength test.

## 1. Introduction

Thermoplastic polyurethane (TPU) is a type of block copolymer that is usually synthesized with a soft segment diol, a hard segment diisocyanate, and a chain extender. The incompatibility between soft and hard segments results in microphase separation, the structure of which dominates the mechanical properties and phase morphology of TPU, which has high intensity, toughness, and wear resistance, as well as the properties of both plastics and elastomers [1,2,3,4]. Through different proportions of soft and hard segments, polyurethane (PU) can be used to synthesize materials with different properties, which can be applied in various fields [5,6]. Therefore, TPU is of great industrial importance [7]. However, compared with other thermoplastic elastomer materials, the poor thermal stability of TPU [8,9] results in limitations in back-end applications.

The thermal properties of TPU can be improved by using blending [10,11,12], copolymers [13], and cross-linked structures [14]. Another approach is the functionalized introduction of fluorine-containing chemicals to form fluoroacrylate polyurethane (FPU), which is expected to have properties similar to those of other fluorinated polymers [15], including biocompatibility, excellent environmental stability, hydrolysis resistance, thermal stability [16,17], chemical resistance, low interface free energy, and water and oil resistance [18]. The interaction of organic fluorine is known to involve π–πF, C–F∙∙∙H, F∙∙∙F, C–F∙∙∙πF, C–F∙∙∙π, C–F∙∙∙M+, C–F∙∙∙C=O, and anion–πF [19]. Therefore, after fluorine is introduced into TPU to form fluoroacrylate thermoplastic polyurethane (FTPU) [20], the stronger interaction of the C–F∙∙∙H hydrogen bond (HB) compared with that of the C=O∙∙∙H HB, together with the interactions of other organic fluorine enable FTPU to possess stronger molecular interactions, resulting in higher rigidity. Moreover, the –CF_2_– group of organic fluorine coats the C–C bond with fluorine atoms, which effectively enhances the chemical resistance, barrier properties, and thermal stability of FTPU, leading to wider applications. Liu et al. [21] mixed fluorinated polyether diol with polybutylene adipate in different proportions to produce soft segments, which were then combined with MDI to prepare FTPU. Studies have shown that fluorinated chain extenders with low reactivity affect the final molecular weight of FTPU and can improve thermal stability. Yang et al. [22] successfully introduced 4,4′-[2,2,2-trifluoro-1-(trifluoromethyl)ethylidene] bisphenol into TPU to form FTPU, and their results showed that introducing fluorine-containing chain extenders improved the thermal stability and rigidity of TPU. Furthermore, our laboratory [23,24] successfully completed synthesis and introduced short-segment fluorine-containing chain extenders into the main and side chains of TPU, which enhanced thermal properties, tensile strength, and hydrophobicity. The above studies indicated that fluorine-containing chain extenders can effectively improve the thermal stability and mechanical properties of PU. Moreover, it was found that if a short-segment fluorine-containing chain extender was introduced as a side chain, it was able to effectively increase the tensile strength of FTPU, while the tensile strain and heat stability were lower. If it was introduced as a main chain, it was able to effectively increase the thermal stability and maintain the original breaking strain of PCL and slightly increase the tensile strength. Therefore, in this study, a long-segment fluorine-containing chain extender was introduced and used together with the urethane group to increase molecular interactions, thereby effectively increasing the thermal stability and mechanical properties of FTPU. So, it is hoped to effectively increase the thermal stability and mechanical property at the same time.

In the present study, a novel FTPU was successfully synthesized through the reaction of 4,4’-diphenylmethane diisocyanate (MDI) and polycaprolactone diol (PCL), which formed a PU prepolymer. Subsequently, 1H,1H,10H,10H-perfluor-1,10-decanediol (PFD) was added to the prepolymer to obtain PFD/PU polymers, and their structures were confirmed through Fourier transform infrared spectroscopy (FTIR) and nuclear magnetic resonance (NMR). The resulting PFD/PUs were then subjected to molecular weight determination by using varying amounts of PCL and PFD chain extenders, and the physical and chemical properties of the PFD/PUs were investigated.

## 2. Experimental

### 2.1. Materials

1H,1H,10H,10H-Perfluor-1,10-decanediol (PFD) was purchased from Matrix Scientific (Columbia, SC, USA). 4,4’-Diphenylmethane diisocyanate (MDI), polycaprolactone diol (PCL, Mw = 530), and dibutyltin dilaurate (DBTDL) were purchased from Aldrich (St. Louis, MO, USA). *N*,*N*-Dimethylacetamide (DMAc) was obtained from Mallinckrodt Chemicals (Dublin, Ireland).

### 2.2. Synthesis of PFD/PUs

A 2-step process was used to polymerize the PFD/PUs. First, MDI, PCL, and DMAc were added to a 500 mL 4-neck reaction flask and heated to 80 °C using a heating mantle. After 2 to 3 drops of dibutyltin dilaurate were added, the solution was mixed using a mechanical stirrer at 200 rpm, and PU prepolymers were formed after 2 h of reaction. In the second step, the PFD chain extenders were dissolved in DMAc and slowly dripped into the reaction flask, and the reaction was continued for 2 h (Scheme 1). In the polymerization, the di-n-butylamine method [25] was used to calculate the NCO content in all steps to monitor the reaction process. The obtained PFD/PU solution was subjected to vacuum defoaming for 2 h, after which it was poured into a serum bottle and stored in a refrigerator for 1 day. Finally, the PFD/PU solution was poured into a Teflon plate and dried in a temperature-programmable circulating oven for 8 h. The recipe, symbols, and theoretical contents of the hard and soft segments for the PFD/PU films are shown in Table 1. This study was only concerned with the relative contents of hard (or soft) segments of PFD/PUs with different PFD contents, so the theoretical hard (or soft) segment content is sufficient. Therefore, theoretical hard and soft segment contents were calculated by using Equations (1) and (2), respectively, as used in the general literature [26].
(1)$$\mathrm{Theoretical}\;\mathrm{hard}\;\mathrm{segment}\;\mathrm{content}\;(\mathrm{wt}\%)=\frac{W_{\mathrm{MDI}}+W_{\mathrm{PFD}}}{W_{\mathrm{MDI}}+W_{\mathrm{PCL}}+W_{\mathrm{PFD}}}\times 100\%$$

Theoretical soft segment content (wt%) = 100% − Theoretical hard segment content (wt%)
(2)

*W_MDI_*, weight of MDI; *W_PCL_*, weight of PCL; *W_PFD_*, weight of PFD.

### 2.3. Advanced Polymer Chromatography System (APC)

The molecular weights of PFD/PUs were characterized using an Acquity APC core system (Waters Corp., Milford, MA, USA) with Tetrahydrofuran (THF) as eluent at a flow rate of 0.8 mL/min. The measurement was carried out at 45 °C.

### 2.4. Fourier Transform Infrared Spectroscopy (FT-IR)

Fourier transform infrared spectroscopy measurements were performed on a PerkinElmer Spectrum One spectrometer (Waltham, MA, USA). The spectra of the samples were obtained by averaging 16 scans in a range of 4000 to 650 cm^−1^ with a resolution of 2 cm^−1^.

### 2.5. ^1^H NMR Spectrometer

^1^H NMR (in DMSO-d_6_) spectra of the specimens were measured using a Bruker Avance 300 spectrometer (300 MHz; Bruker, Billerica, MA, USA).

### 2.6. ^19^F NMR Spectrometer

^19^F nuclear magnetic resonance (NMR) spectra of the polymers were recorded on a Bruker AVIII HD400 Hz spectrometer (Bruker, Billerica, MA, USA) using DMSO-d_6_ as a solvent and tetramethylsilane as an internal standard.

### 2.7. X-Ray Photoelectron Spectroscopy (XPS)

X-ray photoelectron spectroscopy (XPS) measurements were carried out using a Thermo Fisher Scientific (VGS) spectrometer (Waltham, MA, USA). An Al Kα anode was used as the x-ray source (1486.6 eV), and a binding energy range of 0 to 1400 eV was selected for the analysis. The binding energies were calibrated to the C1s internal standard with a peak at 284.8 eV. The high-resolution C1s spectra were decomposed by fitting a Gaussian function to an experimental curve using a nonlinear regression.

### 2.8. Surface Roughness Analysis

Scanning was performed using a CSPM5500 atomic force microscope from Being Nano-Instruments (Beijing, China), which is generally operated in 2 imaging modes: tapping and contact. The tapping mode was used in this study, and the tip of the oscillation probe cantilever made only intermittent contact with the sample. Regarding the phase of the sine wave that drives the cantilever, the phase of the tip oscillation is extremely sensitive to various sample surface characteristics; therefore, the topography and phase images of a sample’s surface can be detected.

### 2.9. Thermogravimetric Analysis (TGA)

Thermogravimetric analysis was performed on a PerkinElmer Pyris 1 TGA (Perkin Elmer, Waltham, MA, USA). The samples (5–8 mg) were heated from room temperature to 700 °C under nitrogen at a rate of 10 °C/min.

### 2.10. Differential Scanning Calorimetry (DSC)

Differential scanning calorimetry was performed on a PerkinElmer Jade differential scanning calorimeter (Perkin Elmer, Waltham, MA, USA). The samples were sealed in aluminum pans with a perforated lid. The scans (−50 to 50 °C) were performed at a heating rate of 10 °C/min under nitrogen purging. The glass transition temperatures (T_g_) were located as the midpoints of the sharp descent regions in the recorded curves. The melting points were recorded as the peak maximum of the endothermic transition in the second scan. Approximately 5–8 mg of samples were used in all of the tests.

### 2.11. Dynamic Mechanical Analysis (DMA)

Dynamic mechanical analysis was performed on a Seiko dynamic mechanical spectrometer (model DMS6100) at 1 Hz with a 5 μm amplitude over a temperature range of −50 to 50 °C at a heating rate of 3 °C/min. DMA was conducted in tension mode with specimen dimensions of 20 mm × 5 mm × 0.2 mm (L × W × H). The T_g_ was taken as the peak temperature of the glass transition region in the tan δ curve.

### 2.12. Stress–Strain Testing

Tensile strength and elongation at break were measured using a universal testing machine (MTS QTEST5, model QC505B1, MTS Sys. Corp., Cary, NC, USA). Testing was conducted with ASTM D638. The dimensions of the film specimen were 45 mm × 8 mm × 0.2 mm. Every spectrum was tested 3 times and the average value was obtained.

## 3. Results and Discussion

### 3.1. Gel Permeation Chromatography Analysis

Figure 1 shows the gel permeation chromatography curves for PFD/PUs synthesized under different PFD proportions. As shown, the weight distributions of the PFD/PUs were unimodal, revealing that the synthesis was complete and without material residues. The molecular weight data are presented in Table 2. The results show that a high PFD content increased the effluent time, and the value of the molecular weight distribution (Mw/Mn; dispersity index) calculated using the PFD/PUs fell within 1.6–1.8. A decline in the viscosity of the PFD/PU polymer solution following the increase in PFD content also occurred during the synthesis. These results reveal that increasing the PFD chain extender content reduced the molecular weight of the FTPU. This was because the carbon chain number and molecular weight of PFD are higher than those of other chain extenders, such as 1,4-butanediol and ethylene glycol used in general PU. Therefore, the reduced molecular weight of the PFD/PUs following an increase in PFD was probably due to the effects of activity. The trend was the same as that in the study by Yang et al. [22], in which an increase in fluorine content reduced the molecular weight.

### 3.2. FTIR

Figure 2a shows the FTIR spectrum of the PFD/PUs at a wavenumber range of 4000–650 cm^−1^. The spectrum revealed that the polymers had five major common peaks: –NH stretching vibration peak (3333 cm^−1^), CH_2_ stretching vibration peak (2925 and 2862 cm^−1^), C=O (amide I band, near 1727 cm^−1^), –NH (amide II band, 1534.68 cm^−1^), stretching vibration peak of the C–F group (1221–1205 cm^−1^), and C–O stretching vibration peak (near 1098–1071 cm^−1^). Moreover, no free NCO group was observed at 2240–2275 cm^−1^. Therefore, MDI had fully reacted with the PCL or PFD chain extender during the synthesis processes and the yields of PFD/PUs were all 100%.

Figure 2b shows the absorption peak within the wavenumber range of 1900–1000 cm^−1^. FTIR analysis conducted by Yang et al. [27] showed that C=O functional groups obtained from the PU system using the curve-fitting technique included C=O_free_, C=O_HB disordered_, and C=O_HB ordered_, with C=O_HB ordered_ appearing at approximately 1724, 1701, and 1660 cm^−1^. Wang et al. [28] subjected FPU to an FTIR test, and the data indicated that when the C=O and –NH functional groups appeared at three peaks (free, disordered, and ordered) in the curve fitting, C–O and C–F functional groups produced two peak values (free and HB). The HB percentage of FPU was calculated at 1530 cm^−1^ because the stretching vibration peak of the benzene ring at this wavelength did not overlap with the other peak values. Therefore, the existence of HBs in the PU was proved using the following formula:(3)$$A\%=\frac{I_{H}/I_{\mathrm{ref}}}{I_{H}/I_{\mathrm{ref}}+I_{\mathrm{free}}/I_{\mathrm{ref}}}$$
where I_H_ is the HB strength, I_free_ is the free radical bonding strength, and I_ref_ represents the absorption intensity at 1534 cm^−1^. Additionally, according to the experimental data in this study, the characteristic peaks of the N–H, C=O, C–F, and C–O groups affected by the HBs also appeared in the PFD/PUs, with H–bonded C=O, H–bonded C–F, and H–bonded C–O located at 1646, 1205, and 1098 cm^−1^, respectively. This confirmed that a weak HB existed between N–H and C–F.

Figure 3 shows the absorption peaks within the wavenumber range 1240–1190 cm^−1^. As shown, the three peak values that appeared at 1240–1190 cm^−1^ were amide III, C–F_free_, and C–F_HB_. The HB percentage of N–H∙∙∙F–C was calculated using Equation (3), with PFD/PU-01, PFD/PU-02, and PFD/PU-03 having an HB percentage of 23.63%, 27.41%, and 31.18%, respectively. Accordingly, an increase in PFD enhanced the HB interaction in the FTPU film.

### 3.3. Fluorine-19 NMR

Figure 4 shows the molecular structure of the fluorine parts of the PFD/PUs and the ^19^F NMR analytical chart for PFD/PU-01. The figure shows three absorption peaks, labeled 1–3. ^19^F–^19^F COSY of PFD/PU-01 was performed to accurately analyze F1–F3. Figure 5 shows two signals of F2 (labeled F2 and F2′) and three strong correlations (F2–F4, F2–F2′, and F1–F2′). Fluorine spectrum studies have found that ^4^J(F,F) is stronger than ^3^J(F,F) [29], indicating that a strong coupling exists between the next signals (i.e., ^4^J(F,F)). Figure 5 also shows that F–A and F–A’ were the most affected by the other elements and were thus labeled F1. In a similar fashion, the peaks at −119.78 ppm (F1), −121.59 ppm (F2 and F2′), and −123.83 ppm (F3) corresponded to F–A and F–A’; F–C, F–C’, F–D, and F–D’; and F–B and F–B’, respectively. The corresponding positions of fluorine were confirmed, as shown in Figure 6. Such coupling was insufficient to verify that the fluorinated chain extender was attached to the PU, and additional identification through 2D NMR spectroscopy (^1^H–^19^F COSY, ^1^H–^13^C HMBC) was required. Figure 7 shows the ^1^H–^19^F COSY diagram for the PFD/PUs. The spectrum revealed that the H atoms of the CH_2_ group in the fluorinated chain extender were located at 4.89 ppm and had a relevant coupling (^3^J(H,F)) with F1. Additionally, Figure 7 illustrates that the H atoms at 4.89 ppm shared a weak coupling (^4^J(F,F)) with F3, which again verified that the analysis was correct. Figure 8 shows the ^1^H–^13^C heteronuclear multiple bond correlation (HMBC) spectrum of PFD/PU-01. According to the literature, the PU ester O–C=O is located at approximately 153 ppm [30], which revealed that the C=O location corresponded to the location of the H atoms of CH_2_ (4.89 ppm). The aforementioned analysis showed that the PFD chain extender had successfully reacted with MDI to form urethane groups.

### 3.4. X-Ray Photoelectron Spectroscopy

Figure 9 shows XPS spectra for PFD/PUs, with each spectrum containing four main peaks: C1s, O1s, N1s, and F1s. The element composition and peak-related properties are listed in Table 3. These findings revealed that the binding energies of C1s, O1s, N1s, and F1s decreased following an increase in PFD content, and the F content increased from 2.31% to 9.47%. Compared with PFD/PU-02 and PFD/PU-03, the F1s binding energies of the C–F bond in PFD/PU-01 exhibited a clear offset from 690 to 688 eV. Accordingly, the molecular interaction in the PFD/PU film changed when the PFD content increased. The O1 elements in PFD/PU-01 and PFD/PU-03 were subjected to an XPS peak-differentiation-imitating analysis (Figure 10). As shown, the O–C=O* binding energies of PFD/PU-01 and PFD/PU-03 were 532.08 and 531.98 eV, respectively. Berger et al. [19] concluded that the interaction of organic fluorine reveals that a dipole–dipole interaction exists between C–F∙∙∙C=O. These results reveal that the lower transfer values of the C1s, O1s, N1s, and F1s binding energies confirmed the interaction between the –C=O group and C–F in the PFD/PUs [31].

Figure 11 shows the XPS peak-differentiation-imitating analysis of C1s plotted for the PFD/PUs with different proportions of the PFD chain extender. The C–C binding energy distributed by the C1s curve in the PFD/PUs was approximately 285.0 eV, and approximately 286 eV for C–O, 287 eV for C–O–C, and 292 eV for C–F_2_. The corresponding peak produced by O–C=O was approximately 288 eV [32], and it was distributed to the carbonyl group in the urethane group. Table 3 illustrates that the nitrogen content increased following an increase in PFD and that the amount of nitrogen represented the amount of hard segments. Substantial HB interactions between C–F and N–H may also have existed within the PFD/PUs; thus, fluorine chains were believed to facilitate the pulling of the hard segments to the surface of the PU [33]. According to the figure, the increased PFD content led to a shift in the C–F position from 292 to 293.0 eV, a change in the peak intensity, and an increase in C–N binding energy from 285.88 to 285.95 eV. The reason was that increasing the PFD increased the number of C–F∙∙∙H–N HBs, which in turn increased the C–F binding energy, a result that was consistent with the FTIR analysis. Moreover, the binding energies of C–O, C–O–C, and O–C=O decreased following an increase in PFD content. This may have been caused by introducing long-chain fluothane segments into the PU, which disrupted the original HB reaction of the PU due to a steric hindrance, thereby reducing the binding energy. However, C–F∙∙∙H–N had a greater HB interaction than did C=O∙∙∙H–N because of the high electronegativity of fluorine, and a stronger HB interaction was produced in the PU film.

### 3.5. Surface Roughness Analysis

The left and right images in Figure 12a–c show the topography and phase data images for PFD/PU-01, PFD/PU-02, and PFD/PU-03, respectively. The PFD/PUs exhibited some continuous protrusions in the topography. The average surface roughness of PFD/PU-01, PFD/PU-02, and PFD/PU-03 was 2.17, 2.72, and 4.45 nm, respectively. The results revealed that the surface roughness increased when the PFD content increased, which caused a rougher FTPU. This phenomenon was attributed to the increase in the hard segments of the FTPU and the interaction between CF_2_ and C=O in the hard segments following the increase in PFD. In other words, increasing the PFD chain extender increased the HB interaction in the FTPU film, which in turn caused aggregations or protrusions on the film’s surface [24]. Additionally, numerous continuous irregular granular and stripe phases were observed in the phase diagram of the PFD/PUs, which increased as the PFD content increased. These irregular phases revealed that the hard segments were rich in PFD chain extender [34,35], a phenomenon that was consistent with the findings in the XPS spectrum.

### 3.6. Thermal Properties

Figure 13 illustrates the thermogravimetric analysis (TGA) curve of the PFD/PUs synthesized with different amounts of PFD chain extender. The initial decomposition temperature of the PFD/PUs was defined as T_onset_, which related to pyrolysis of the FTPU. The data revealed that the T_onset_ of PFD/PU-01, PFD/PU-02, and PFD/PU-03 was 299.2, 305.1, and 308.6 °C, respectively; the thermogravimetric data of PFD/PUs are presented in Table 4. The results show that T_onset_ increased when the PFD chain extender content increased in the PFD/PUs. This could be attributed to the strong bonding energy of –CF_2_ (540 kJ/mol), which required relatively high energy to break the bond. Furthermore, the covalent radius of the fluorine atom was equivalent to half the C–C bond length; thus, fluorine atoms shielded the main C–C chain and ensured its stability. Moreover, the interaction between the C=O and –CF_2_ groups was verified through FTIR, and the polar bonding of –CF_2_ contributed to the formation of the phase separation of hard segments of the PU film in the soft segment [36]. The additional PFD increased the thermal stability of the PU film. The residual weight at 700 °C exhibited an increase when the PFD increased, which consequently reduced the amount of PCL that was required. This resulted in more hard segments, which facilitated carbon formation.

Figure 14 shows the differential scanning calorimetry thermograms of the PFD/PUs with different PFD content, and the relevant data are displayed in Table 4. The results reveal that the glass transition temperature (T_g_) points of PFD/PU-01, PFD/PU-02, and PFD/PU-03 were 3.7, 5.6, and 10.3 °C, respectively. T_g_ was related to the soft segment that consisted of repeat linkages of reacted alternative MDI and PCL units. Previous FTIR spectra indicated the presence of a strong interaction between the C=O groups in soft segments and the –CF_2_ groups in hard segments of the PFD/PUs. When PFD content was higher, more –CF_2_ groups therein would cause a higher interaction that inhibited the segmental chain motion in the PFD/PUs, consequently increasing the T_g_ of PFD/PUs. In other words, the PFD/PUs with more hard segments or chain extenders would have higher T_g_ as previously reported [37].

### 3.7. Dynamic Mechanical Analysis

Figure 15 shows the tan δ and loss modulus (E’’) of the PFD/PUs with different amounts of PFD chain extenders. The dynamic T_g_ was defined as T_gd_. As shown, the T_gd_ of PFD/PU-01, PFD/PU-02, and PFD/PU-0 from the tan δ curve was 7.9, 10.8, and 13.3 °C, respectively, whereas the T_gd_ from the E’’ curves was 0.1, 3.6, and 5.4 °C, respectively. The T_g_ values obtained using different testing methods are listed in Table 5. The results show that the T_gd_ of the PFD/PUs increased as the PFD chain extender content increased. This may have been caused by the inhibition of segmental motion of the PFD/PUs following the increase in hard segments and C–F∙∙∙H–N HB interactions that increased the T_gd_ of the PFD/PUs, a finding that was similar to the results from the thermal property analysis described previously. The tan δ curves of the PFD/PUs indicate a decrease in tan δ_max_ with an increase in PFD content. This was because of the increased hard segments and influence of the HB interactions following the increased PFD, resulting in more elastic PFD/PUs, because the value of tan δ was obtained by dividing the value of E’’ by E’. Therefore, PFD/PU-03, which had the highest fluorine content, showed the lowest peak value. In other words, the hard segments containing PFD units were harder than the soft segments containing PCL units. Additionally, the dipole–dipole interaction between C–F∙∙∙C=O contributed to the blocking of the segmental activity of the PFD/PUs. In summary, the PFD/PUs with relatively high PFD content were elastic, which suggests that increasing the PFD content improves the rigidity of PFD/PUs.

### 3.8. Tensile Properties

Figure 16 shows the stress–strain curves of the PFD/PUs synthesized with different amounts of PFD chain extender, and the data on their mechanical properties are listed in Table 6. According to the results, PFD/PU-01, PFD/PU-02, and PFD/PU-03 had, respectively, a maximum tensile strength of 8.23, 16.34, and 21.55 MPa; an extension at break of 1630%, 1434%, and 1156%; and a Young’s modulus of 0.5, 1.4, and 2.1 MPa. These results reveal that increasing the PFD content increased the tensile strength and Young’s modulus. This is due to the following: first, the FTPU film was more rigid when FTPU contained more PFD or hard segments; and second, the HBs produced between –NH and CF_2_ in the PFD/PUs inhibited the segmental motion of the PFD/PUs. The result was an increase in tensile strength and Young’s modulus in the FTPU film. The results of the mechanical property curve are consistent with those of the dynamic mechanical analysis.

## 4. Conclusions

In this study, PFD was introduced into PU to produce FTPU, and ^1^H NMR, ^19^F NMR, ^19^F–^19^F COSY, ^1^H–^19^F COSY, and HMBC confirmed the successful synthesis of PFD/PUs. The results from FTIR and XPS indicate that introducing the CF_2_ group into the PFD produced an HB interaction with the –NH group; a high PFD content resulted in more HB interactions in the PFD/PUs. AFM showed that PFD contributed to the microphase separation of the PFD/PUs because of the HB interactions. Thermal property analysis showed that because of the strong binding energy of –CF_2_ in the PFD (540 kJ/mol) and because the covalent radius of fluorine atoms is equivalent to half the C–C bond length, increasing the PFD content shielded the main C–C chain and enhanced the thermal stability of the PFD/PUs. Dynamic mechanical analysis and tensile strength tests also confirmed that the PFD extender enhanced the rigidity of the PU film.

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
