# Peer review of "Synthesis and Properties of Novel Polyurethanes Containing Long-Segment Fluorinated Chain Extenders"

_polymers, 2018, doi:10.3390/polym10111292_

Reviewer 1 Report

In the article “synthesis and properties of novel biodegradable polyurethanes containing long-segment fluorinated chain extenders” the authors describe the synthesis and present a deep characterization of the synthesized polyurethanes with a fluorinated chain extender. In this ways, I consider that is a suitable article for being accepted after some minor corrections:

-          First, authors mention in the title that these polyurethanes are biodegradable, but they have not present any characterization about it. Thereby, I think that authors should carried out some tests or by contrast modify the title.

-          Authors comment that the prepopylmerization was carried out in two hours, but have authors verified by any method that the prepolymer reaction occurred completely? In the same way, how was determined the second step of the reaction?

-          Authors mention that the films presented a Tg around 3-11 °C and in lines 317 and 318 explained that films exhibit a greater phase separation with the increase of HS content. What can authors explain about that Tg? Is that Tg related with the SS or HS? In the case that it was related with the SS, it would not be possible the fact of Tg value increase with the increase of phase separation because the tg of the pure PCL is lower. Please, I would be grateful if you could explain more deeply that transition.

Author Response

Manuscript ID:polymers-389247

Title:Synthesis and properties of novel biodegradable polyurethanes containing long-segment fluorinated chain extenders

Comments 1.

In the article “synthesis and properties of novel biodegradable polyurethanes containing long-segment fluorinated chain extenders” the authors describe the synthesis and present a deep characterization of the synthesized polyurethanes with a fluorinated chain extender. In this ways, I consider that is a suitable article for being accepted after some minor corrections:

1.     First, authors mention in the title that these polyurethanes are biodegradable, but they have not present any characterization about it. Thereby, I think that authors should carried out some tests or by contrast modify the title.

Ans:The title of the manuscript have been revised of the revised manuscript. (Synthesis and properties of novel polyurethanes containing long-segment fluorinated chain extenders)

2.     Authors comment that the prepopylmerization was carried out in two hours, but have authors verified by any method that the prepolymer reaction occurred completely? In the same way, how was determined the second step of the reaction?

Ans: In the polymerization, the di-n-butylamine method was used to calculate the NCO content in all steps to monitor the reaction process and the statement has been added in line 92-93 of the revised manuscript.

3.     Authors mention that the films presented a Tg around 3-11 °C and in lines 317 and 318 explained that films exhibit a greater phase separation with the increase of HS content. What can authors explain about that Tg? Is that Tg related with the SS or HS? In the case that it was related with the SS, it would not be possible the fact of Tg value increase with the increase of phase separation because the tg of the pure PCL is lower. Please, I would be grateful if you could explain more deeply that transition.

Ans: The statements in lines 317 and 318 of the original manuscript only claimed that films exhibit a greater phase separation with the increase of HS content [37]. There was no description about Tg or the relationship between Tg value and the phase separation degree in that paragraph. The relationship between Tg value and the HS content was stated in the next paragraph and has been explained more deeply in lines 338-344 of the revised manuscript.

Comments 2.

1.     The major objective of this research is the synthesis and characterization of  fluorinated polyurethanes to improve the thermal properties. The authors used well known starting materials (commercially available materials) and the preparation methods are also commonly used in the PUR chemistry.

Ans:Thank the reviewer for his suggestion.

2.     Since the biodegradability is not investigated and proved the title is misguiding. It should be changed by removing the “biodegradable “ term.

Ans:The title of the manuscript have been revised of the revised manuscript. (Synthesis and properties of novel polyurethanes containing long-segment fluorinated chain extenders)

3.     In the introduction the authors should clearly show what the research group has been done before, and what is the progress to compare with them.

Ans: But it was found that if a short-segment fluorine-containing chain extender was introduced as a side chain, it was able to effectively increase the tensile strength of FTPU, while the tensile strain and heat stability were lower. If it was introduced as a main chain, it was able to effectively increase the thermal stability and maintain the original breaking strain of PCL and slightly increase the tensile strength. Therefore, in this study, a long-segment fluorine-containing chain extender was introduced and used together with the urethane group to increase molecular interactions, thereby effectively increasing the thermal stability and mechanical properties of FTPU. So, it is hoped to effectively increase the thermal stability and mechanical property at the same time. The statement has been added in line 65-72 of the revised manuscript.

4.     The use of abbreviations is confusing, and some mistakes are also occurred, therefore careful checking is necessary.

Ans:The abbreviation part in the text was carefully reexamined and revised.

5.     From the description of synthesis of PFD/PUs the yields are missing.

Ans: The yields of PFD/PUs are all 100% as evidenced by the FTIR spectra and the statement has been added in line 188 of the revised manuscript.

6.     The legend of Table1. is not informatvie enough and misguiding. The hard and soft segment contents of the block copolymers can not be calculated by using eqs.2.1 and 2.2. In this way only the feed composition can be obtained which is regularly not identical with the coposition of the block copolyner.  Independent quantitative NMR investigation is necessary.

Ans: The legend of Table 1 has been revised in line 104 of the revised manuscript. Indeed, the actual hard and soft segment contents of the block copolymers cannot be calculated by using eqs.2.1 and 2.2 and quantitative NMR investigation is necessary. However in this paper, only the relative contents of hard (or soft) segment of PFD/PUs with different PFD contents were concerned, so the theoretical hard (or soft) segment content is sufficient. Therefore theoretical hard and soft segment contents were calculated by using eqs.2.1 and 2.2 respectively, as were adopted in general literatures. The related corrections have been revised in lines 96-107 of the revised manuscript.

7.     In Fig.1 the full GPC traces should be shown to support the statement in lines 154 and 155.

Ans: The Fig. 1 have been revised in the revised manuscript.

8.     There is an empty sentence in lines 165 and 166.

Ans: Moreover, the molecular weight and dispersity index value in this study were within a reasonable range and did not affect the material properties. The related statement has been delated of the revised manuscript.

9.     The sections of 3.3 and 3.4. should be shortened.

Ans: The section 3.3 and 3.4 have been shortened of the revised manuscript.

Comments 3.

This is a well-written manuscript on the synthesis of fluorine-containing polyurethane. The system is extensively characterized. I have only one minor comments: please indicate the standard deviation of the tensile properties in Table 6. Also, please indicate the elongation rate of stress train testing and the number of replicates of this test.

Ans: Every spectrum was tested for three times and the average value was obtained and the statement has been added in line 158-159 of the revised manuscript. The table 6 have been revised of the revised manuscript.

Reviewer 2 Report

1.       The major objective of this research is the synthesis and characterization of  fluorinated  polyurethanes to improve the thermal properties. The authors used well known starting materials (commercially available materials) and the preparation methods are also commonly used in the PUR chemistry.

2.       Since the biodegradability is not investigated and proved  the  title is misguiding. It should be changed by removing  the “biodegradable “ term.

3.       In the introduction the authors should clearly show  what the research group has been done before, and what is  the progress to compare with them.

4.       The use of abbreviations  is confusing,  and some mistakes are also occurred, therefore careful checking is necessary.

5.       From the description of synthesis of PFD/PUs the yields are missing.

6.       The legend of Table1. is not informatvie enough and  misguiding. The hard  and soft segment contents  of the block copolymers can not be calculated by using eqs.2.1 and 2.2. In this way only the feed composition  can be obtained  which is regularly not identical with the coposition of the block copolyner.  Independent quantitative  NMR  investigation is necessary.

7.       In Fig.1 the full GPC traces  should be shown to support the statement  in lines 154 and 155.

8.       There is an empty sentence in lines165 and 166.

9.       The sections of 3.3 and 3.4. should be shortened.

Author Response

(The authors gave the same response as above.)

Reviewer 3 Report

This is a well-written manuscript on the synthesis of fluorine-containing polyurethane. The system is extensively characterized. I have only one minor comments: please indicate the standard deviation of the tensile properties in Table 6. Also, please indicate the elongation rate of stress train testing and the number of replicates of this test. 

Author Response

(The authors gave the same response as above.)
